# Interplay Effect of Splenic Motion for Total Lymphoid Irradiation in Pediatric Proton Therapy

**DOI:** 10.3390/cancers15215161

**Published:** 2023-10-26

**Authors:** Ozgur Ates, Jinsoo Uh, Fakhriddin Pirlepesov, Chia-ho Hua, Brandon Triplett, Amr Qudeimat, Akshay Sharma, Thomas E. Merchant, John T. Lucas

**Affiliations:** St. Jude Children’s Research Hospital, Memphis, TN 38105, USA; jinsoo.uh@stjude.org (J.U.); fakhriddin.pirlepesov@stjude.org (F.P.); chia-ho.hua@stjude.org (C.-h.H.); brandon.triplett@stjude.org (B.T.); amr.qudeimat@stjude.org (A.Q.); akshay.sharma@stjude.org (A.S.); thomas.merchant@stjude.org (T.E.M.); john.lucas@stjude.org (J.T.L.J.)

**Keywords:** interplay effect, total lymphoid irradiation, pediatric proton therapy, splenic motion

## Abstract

**Simple Summary:**

This study addresses the challenge of respiratory motion in pediatric patients undergoing proton therapy for total lymphoid irradiation (TLI), an essential component of reduced-intensity conditioning regimens for hematopoietic cell transplant (HCT) patients. The primary concern is the interplay effect, which can cause deviations from the planned dose during respiratory motion. The research utilized static and 4D CT images from ten patients to simulate the interplay effect and assess its impact on treatment planning. The study concludes that 4D plan evaluation and robust optimization techniques can help address respiratory motion challenges in proton TLI treatments, especially in cases where other motion management strategies like breath-holding are not feasible due to the patient’s young age, the need for anesthesia, or medical conditions. Patient-specific respiratory motion evaluations are crucial to ensure adequate dosimetric coverage in proton therapy for pediatric patients.

**Abstract:**

(1) Background: The most significant cause of an unacceptable deviation from the planned dose during respiratory motion is the interplay effect. We examined the correlation between the magnitude of splenic motion and its impact on plan quality for total lymphoid irradiation (TLI); (2) Methods: Static and 4D CT images from ten patients were used for interplay effect simulations. Patients’ original plans were optimized based on the average CT extracted from the 4D CT and planned with two posterior beams using scenario-based optimization (±3 mm of setup and ±3% of range uncertainty) and gradient matching at the level of mid-spleen. Dynamically accumulated 4D doses (interplay effect dose) were calculated based on the time-dependent delivery sequence of radiation fluence across all phases of the 4D CT. Dose volume parameters for each simulated treatment delivery were evaluated for plan quality; (3) Results: Peak-to-peak splenic motion (≤12 mm) was measured from the 4D CT of ten patients. Interplay effect simulations revealed that the ITV coverage of the spleen remained within the protocol tolerance for splenic motion, ≤8 mm. The D100% coverage for ITV spleen decreased from 95.0% (nominal plan) to 89.3% with 10 mm and 87.2% with 12 mm of splenic motion; (4) Conclusions: 4D plan evaluation and robust optimization may overcome problems associated with respiratory motion in proton TLI treatments. Patient-specific respiratory motion evaluations are essential to confirming adequate dosimetric coverage when proton therapy is utilized.

## 1. Introduction

Proton therapy is a state-of-the-art radiation treatment modality that has shown promising results in the management of various malignancies. Unlike conventional photon-based radiation therapy, which delivers radiation with continuous energy deposition along its path, proton therapy utilizes charged particles to target tumors with greater precision and reduced damage to surrounding healthy tissues. This characteristic makes proton therapy an attractive option for treating tumors in anatomically complex and radiation-sensitive regions, such as lymphoid tissues [1].

One of the challenges in treating certain lymphoid malignancies is the need to include extensive lymph node volumes and specific lymphoid structures throughout the body. Total Lymphoid Irradiation (TLI) has emerged as a potential solution for these cases and has replaced total body irradiation in some applications where less intensive conditioning regimens are required prior to hematopoietic cell transplant (HCT) [2]. TLI delivers radiation to major lymphatic regions to control or eradicate the disease and/or improve graft tolerance prior to transplant, particularly in cases of lymphoma, leukemia, and autoimmune disorders, where the immune system plays a crucial role [3].

Early studies utilizing TLI primarily employed conventional photon-based radiation therapy, which, though effective, had its limitations due to the risk of substantial radiation exposure to surrounding organs [4,5,6]. The advent of proton therapy has brought a new dimension to the application of TLI, offering enhanced dose conformity and sparing of normal tissues, thus potentially improving treatment outcomes, and reducing long-term side effects [7]. Proton therapy’s unique depth–dose characteristics enable a precise deposition of radiation within the targeted tissues, with minimal exit dose beyond the treatment volume. This feature is especially advantageous when treating lymphoid tissues, which are often located near critical structures like the thyroid, heart, lungs, and spinal cord [8]. A group investigated the long-term risk of secondary cancers following Hodgkin lymphoma radiotherapy, encompassing various malignancies affecting the breast, esophagus, heart, liver, lung, pharynx, spinal cord, stomach, thyroid, bones, and soft tissues [9]. Their findings revealed that proton therapy demonstrated a significant reduction in the cumulative risk of secondary cancers by 40% when compared to intensity-modulated radiation therapy (IMRT) and 28% when compared to Volumetric Modulated Arc Therapy (VMAT). Consequently, proton-based TLI holds the promise of minimizing radiation-induced toxicities and enhancing patient quality of life compared to traditional radiation techniques.

Proton-based TLI was proposed as a key component of conditioning for patients with severe aplastic anemia (SAA) receiving haploidentical donor HCT on an institutional therapeutic protocol (HAPSAA) [10]. SAA is a rare and life-threatening bone marrow disorder characterized by a significant reduction in the production of blood cells, including red blood cells, white blood cells, and platelets [11]. While immunosuppressive therapy is frontline, HCT is a potentially curative treatment option for patients who have failed immunosuppressive therapy [12]. Unfortunately, the potential donor pool may be limited, and many patients will not have a matched sibling donor available to facilitate transplantation. HAPSAA was initiated to provide transplantation options for patients with a limited donor pool. While 50–60% of Caucasian patients have a matched unrelated donor (MUD) available, less than 20% of underrepresented minority patients can find a fully human leukocyte antigen (HLA) MUD in donor registries. Given that all patients will share at least one HLA haplotype with each biological parent, an eligible haploidentical donor can be rapidly identified and acquired for all patients, thus facilitating HCT for conditions like SAA.

Because SAA is a benign curative condition, proton-based TLI was considered as a potential tolerogenic agent given the favorable dose distribution profile and potential to facilitate dose reductions in other cytotoxic chemotherapies with known adverse late effect profiles. Prior to transplantation, the patient undergoes a conditioning regimen, which involves a combination of chemotherapy and/or radiation therapy including proton therapy. The conditioning regimen serves to suppress the patient’s immune system and make space for the incoming donor cells, thereby reducing the risk of graft rejection and engraftment failure [13].

In pencil beam scanning proton therapy, the interplay effect is a phenomenon where target motion during proton beam delivery leads to variations in the dose distribution within the target volume [14]. For this reason, targets approximating areas of active respiratory motion pose a substantial problem. Properly addressing the interplay effect is essential to ensure accurate and effective dose delivery to the target while minimizing potential impacts on nearby healthy tissues. According to a study report [15], the interplay effect was not a concern for pediatric patients with <0.5 mm retroperitoneal tumor motion. However, for children with diaphragmatic tumor motion >10 mm, simulations revealed significant declines in target coverage, up to 48%, due to the interplay effect.

To date, no published studies have reported on treatment planning and motion management strategies for pediatric patients treated with proton-based TLI. In the context of HAPSAA protocol, this research aimed to explore the dosimetric considerations of proton- based TLI in a cohort of ten pediatric patients with a retrospective analysis. This study focused on investigating the influence of the interplay effect on the spleen, the lymphatic system’s largest organ located near the diaphragm, exhibiting dynamic contraction and relaxation during each respiratory cycle.

## 2. Materials and Methods

In the Materials and Methods section, a virtual phantom experiment is explained in Section 2.1. The patient selection and image data are described in Section 2.2. The treatment planning scheme and the interplay effect simulation workflow are presented in Section 2.3 and Section 2.4, respectively.

### 2.1. Virtual Phantom Experiment

A virtual phantom was created using a patient’s CT images to mimic splenic motions at various displacements of ±3 mm, ±6 mm, ±8 mm, and ±10 mm in the inferior to superior direction. To achieve this, a specialized image manipulation software called ImSimQA (version 4.3.2.) (Oncology Systems Limited, Shropshire, UK) was employed. The software created deformation vector fields, which were then applied to produce artificial CT images with precisely known deformations of ±0.1 mm uncertainty. Figure 1 illustrates a patient case with simulated splenic motion with an associated magnitude of 10 mm ± 0.1 mm in the superior direction, depicted by overlaid deformation vector fields on CT images before Figure 1A and after Figure 1B manipulation. To achieve this deformation, a set of 10 anchor points was strategically placed to deform the upper abdomen region, including the diaphragm, spleen, and liver, by known amounts.

After a series of virtual phantom images were produced, an original proton-based TLI plan was optimized on the patient’s nominal CT image based on the Eclipse treatment planning system (TPS) (Varian Medical Systems Inc., Palo Alto, CA, USA). Verification plans were developed by re-calculating the beam spots of the original plan on each deformed image, accounting for predetermined splenic motions. The plan qualities of verification plans were evaluated by comparing their dose distributions and volume parameters with those of the original plan.

### 2.2. The Patient Selection and Image Data

Ten pediatric patients with mediastinal targets who had previously undergone a course of proton therapy and 4D CT scans were randomly selected for evaluation of splenic motion in this retrospective study. The inclusion criteria involved the selection of pediatric patients who adhered to the age limit recommended by the HAPSAA protocol, which defines patients as those up to 21 years old. Furthermore, the random selection process required patients to possess 4D CT scans encompassing both thoracic and abdominal regions within the scan’s field of view (FOV). Exclusion criteria comprised patients falling outside the specified age range, those with insufficient 4D CT scans, or individuals with contraindications that could potentially affect the assessment of splenic motion, such as the presence of motion artifacts in the 4D CT scans. Institutional Review Board (IRB) approval was obtained prior to analysis. The imaging data were collected on the patients using a Philips Vereos PET/CT system (Philips Healthcare, Cleveland, OH, USA) with 120 kVp, 0.06 spiral pitch factor, 0.625 mm collimation, a 50 cm FOV, and slice thickness of 3 mm. The corresponding respiratory cycles were captured via pneumatic bellows belt signals. The number of 10 respiratory phases were sampled by the phase-binning technique for 4D CT reconstruction.

The peak-to-peak splenic motion was measured in the inferior-to-superior direction from the coronal plane of the 4D CT with a measuring tool in the MIM clinical software (Version 7.3.2) (MIM Software Inc., Cleveland, OH, USA). Figure 2 demonstrates the age of the pediatric patients and the magnitude of peak-to-peak splenic motions measured from the corresponding 4D CT images. The findings indicated a weak correlation (R^2^ = 0.18) between patients’ age and splenic motion, as evidenced by the application of linear regression to the dataset.

### 2.3. Treatment Planning Scheme

In this study, the average CT images were extracted from the 4D CT of ten patients using the mean intensity projection technique in the MIM software and exported into Eclipse TPS to build original treatment plans following the guidelines of the HAPSAA protocol. According to the protocol, the proton-based TLI conditioning regimen was prescribed to 8 Gy-RBE in four fractions at 2 Gy-RBE per fraction. The internal target volume (ITV) spleen was recommended to receive D100% = 95% of the prescribed dose, with a goal of ±5% dose variation.

The PROBEAT-V proton therapy system (Hitachi America, Ltd., Santa Clara, CA, USA) was utilized to administer intensity modulated proton therapy (IMPT) treatment plans for ten patients. A class solution was applied to all patients such that the beam arrangements comprised two posterior beams to treat total lymph nodes including spleen with a plan goal of TLI ITV D95% = 95% ± 5%. Range and setup uncertainties were used as ±3% and ±3 mm, respectively, for the robust plan optimization. The TLI ITV was separated into two targets, namely TLI Upper and TLI Lower, having individual isocenters for upper and lower fields with an overlap at the level of mid-spleen for gradient matching. Field robustness was achieved with ±5 mm uncertainty for lower field in the inferior-to- superior direction. Figure 3 highlights the 3D TLI volume, adjacent structures as organs at risk (OARs), and the dose profile drawn across the spleen for gradient matching.

To mitigate the detrimental impact of internal splenic motion on plan quality, the junction of two fields was strategically chosen at the mid-spleen region. This decision was made to avoid severe dosimetric consequences that could arise from large interplay effects if one field were to cover the spleen alone. Field robustness of ±5 mm was also a safeguard approach to eliminate setup errors between two isocenters for treatments. Overall, treatment planning based on average CT, robustness settings for plan and field, and two-field overlap at the mid-spleen were utilized in this planning regimen of the proton-based TLI treatments.

### 2.4. Interplay Effect Simulation Workflow

Dynamic 4D dose accumulation methods to estimate the interplay effect consider the time-dependent delivery sequence along with representative anatomic motion, determined through techniques like 4D CT or 4D MRI. This approach is an integral part of 4D treatment planning, which complements IMPT planning. By incorporating a 4D image dataset, such as 4D CT, and considering the timing of treatment delivery, this method enables the calculation and evaluation of the dose received by moving targets and OARs.

The interplay effect simulation procedure followed the subsequent sequential steps:Patient-specific respiration rate and the timing of each 4D CT phase were determined based on the respiratory signal recorded from the bellows sensor during the scan.The timing and duration of each proton spot was extracted from the plan DICOM file.A fractional dynamic 4D dose was calculated:Spots that were delivered at each respiratory phase were identified, for which the phase at the starting of the delivery was randomly selected.Dose was calculated on each phase volume of 4D CT with the identified corresponding spots.The phase-wise dose distributions were accumulated via the transformation determined by the deformable image registration from each phase volume of 4D CT to the planning CT (i.e., the average CT).
Step 3 was repeated to generate the dynamic 4D dose for all four fractions in the treatment.

An in-house, dedicated MATLAB algorithm (Version R2019b) (MathWorks Inc., Natick, MA, USA) was developed and utilized to execute the aforementioned procedure. This process allowed the simulation of the dose distribution considering the interplay effect between the respiratory motion and the delivery of proton therapy for all treatment fractions. Dynamic 4D and original plan dose distributions and dose volume parameters were compared based on HAPSAA protocol.

## 3. Results

The results section includes the outcomes of two sections: Section 3.1, Virtual Phantom Experiment, and Section 3.2, Interplay Effect Simulation.

### 3.1. Virtual Phantom Experiment

In this experimental study, the splenic motion was artificially varied with magnitudes ranging from ±3 mm to ±10 mm in the inferior-to-superior direction. The positive sign represented movement in the superior direction, while the negative sign indicated movement in the inferior direction. To have a standard reference, a nominal TLI plan was created based on the recommendation of HAPSAA protocol. Subsequently, when comparing the verification plans to the nominal plan, differences in plan quality were observed, indicating deviations from the plan goal of ITV spleen, D100% = 95% ± 5%, as shown in Table 1.

It was observed that the plan quality began to deteriorate when the artificial splenic motion exceeded ±6 mm. This motion corresponds to a peak-to-peak splenic motion of 12 mm. Beyond this threshold, the plan quality showed signs of degradation for the moving spleen target.

### 3.2. Interplay Effect Simulation

In examining the relationship between splenic motion and ITV D100% coverage, it was observed that the coverage stayed within protocol tolerance for splenic motion up to 8 mm. However, a noticeable decline in coverage occurred with 10 mm of splenic motion, resulting in a 5.7% reduction. A more substantial reduction of 7.8% in ITV D100% coverage was recorded when splenic motion increased to 12 mm. Table 2 provides a comprehensive overview of the interplay effect simulation involving ten patients. It highlights the impact on plan quality concerning ITV spleen, ITV TLI Lower and Upper targets, and the influence on neighboring normal tissues including the left lung and left kidney, considered as OARs.

The findings indicated a significant inverse relationship (R^2^ = 0.73) between splenic motion and the adverse impact on plan quality, stemming from the considerable interplay effect between target motion and beam delivery. Notably, the spleen coverage was affected more profoundly compared to the normal tissues of the left lung and left kidney due to the stringent dose constraint of D100% for ITV spleen. The mean and standard deviation (SD) of D50% in the original plans for the left lung and left kidney were 3.0 ± 1.7 Gy-RBE and 2.7 ± 2.4 Gy-RBE, respectively. In contrast, the interplay effect simulation plans exhibited slightly elevated values, with a mean and SD of D50% for the left lung dose and left kidney of 3.1 ± 1.7 Gy-RBE and 2.8 ± 2.4 Gy-RBE. This increase in dose was minimal, only reaching 0.1 Gy-RBE for the interplay effect plans.

Figure 4 illustrates the dose subtraction performed between the original and interplay effect plans, depicted across the coronal (Figure 4A) and sagittal (Figure 4B) planes. Additionally, it displays the cumulative (Figure 4C) and differential (Figure 4D) dose volume histograms (DVH) for patient case #9, wherein a splenic motion of 10 mm was measured. While the normal organs of the left lung and left kidney displayed comparable results in the DVH curves for the original and interplay effect simulation plans, ITV spleen exhibited significant variations with noticeable cold and hot regions evident in the lower and higher dose levels depicted in the cumulative DVH curves.

In the differential DVH curves plotted in Figure 4D, the dose distributions within specific volume intervals of ITV spleen are depicted, providing insight into how the dose was distributed across the same ITV spleen volume for both the original and interplay effect plans. The assessment of homogeneity index (HI) and conformity index (CI) indicated that the original plan displayed greater homogeneity (HI = 0.09) and more conformal dose distributions (CI = 0.99). Conversely, the interplay effect plan experienced proton range errors, leading to the presence of both cold and hot regions with a reduced homogeneity (HI = 0.24) and slightly lower conformity (CI = 0.95). Upon evaluating the dose subtraction between the original and interplay effect plans, the analysis revealed a mean dose difference of 0.03 ± 0.21 Gy-RBE within the range of +0.74 and −1.29 Gy-RBE for the ITV spleen volume. Notably, 1% of the volume suffered from at least ±0.5 Gy-RBE dose discrepancy (or 6.25% of Rx = 8 Gy-RBE), resulting in hot and cold regions in the ITV spleen volume.

## 4. Discussion

To the best of our knowledge, this study marks the first report discussing the proton- based TLI conditioning regimen and the dosimetric considerations in dealing with a dynamic motion of spleen as part of TLI treatment. Our investigation reveals that once the splenic motion amplitude surpasses 8 mm in the inferior-to-superior direction, there is a noticeable decrease in target coverage. In such instances, it becomes prudent to explore active motion management strategies, such as implementing repainting techniques or utilizing deep inhalation breath holds [16,17].

Our findings align with prior research publications in this field. In a study quantifying the motion of abdominal organs in pediatric cases [18], it was observed that the liver and spleen exhibited greater motion compared to the kidneys (n = 35). Specifically, the splenic motion from inferior to superior was measured as 6.0 ± 3.5 mm. Among these pediatric cases, 31% displayed splenic motion exceeding 8 mm. The same team also carried out an independent investigation on interplay effect simulation [15]. This study comprised a diverse set of cases involving pediatric mediastinal and abdominal patients (n = 10). Notably, only a single case featured the spleen as the disease site, demonstrating a substantial splenic motion of 13.5 mm. Within this context, the study observed the most significant reduction in spleen target coverage, measuring 48% in V99%.

One research group [19] evaluated target volume motion in pediatric neuroblastoma cases (n = 7), noting a maximum organ motion of 2.7 mm in the inferior-to-superior direction. Their investigation unveiled a decrease of 1.9% in target coverage (V95%) due to this motion. Another research effort [20] highlighted the impact of interplay effects through fractionation in a cohort of nine patients with pancreatic cancer. These patients experienced daily variations in mean motion amplitudes of up to 5.3 mm. After an average of seven treatment fractions (range: 2–14), the study demonstrated a notable averaging effect of the interplay effect. This led to sufficient coverage of CTV within the range of 95–107%.

According to the AAPM’s TG-290 on respiratory motion management for particle therapy [14], proton dose distributions are highly sensitive to density changes along the beam path, often caused by factors like respiration, which can affect the beam range. To tackle this challenge, it was advised to undertake treatment planning based on the average CT scan, thereby mitigating some of these influences. As part of a quality assurance procedure for assessing motion, it was suggested to conduct calculations for worst-case scenarios involving two extreme respiratory phases—maximum inhalation and exhalation.

In our virtual phantom experiment, Section 3.1, we evaluated the impact of splenic motion on plan quality by calculating the original plan dose on artificially deformed CT images with known amounts, mimicking two extreme scenarios as maximum inhalation and exhalation phases. We observed that the initial treatment plan began to deteriorate when facing splenic motion exceeding 12 mm. However, the simulations of the interplay effect, Section 3.2, revealed that beyond 8 mm of splenic motion, the dose deviation was deemed unacceptable. The absence of the beam delivery timing might have contributed to this outcome in the virtual phantom study. The re-calculation of the full fraction dose was conducted solely on a single breathing phase, potentially leading to an underestimation of the plan degradation. This underestimation could be attributed to the omission of interplay effects that arise between beam delivery timing and respiratory motion.

The interplay effect simulation incorporated various assumptions. For instance, the patient’s respiration was modeled as a regular waveform, although some patients may exhibit an extended duty cycle during exhalation or inhalation. Additionally, the allocation of spots during each respiration phase was based on random selection, assuming the possibility of beam delivery commencing at any of the ten breathing phases. Furthermore, the orientation of spot scanning was configured to be perpendicular to the splenic motion, resembling horizontal beam scanning. It is essential to note that the scanning direction has the flexibility to be reconfigured vertically, aligning parallel to the target’s motion.

The interplay effect simulation retains its significance as a preliminary quality assurance measure before commencing treatments. AAPM’s TG-290 [14] recommends the utilization of 4D accumulated dose or 4D dose optimization, both of which involve calculations from each 4D respiratory phase. However, it is important to recognize that 4DD is computed exclusively using the treatment plan and the 4DCT image set, omitting the temporal aspect of delivery fluence. On the other hand, dynamically accumulated 4D dose or dynamic 4D dose incorporates the time-dependent delivery sequence and radiation fluence, alongside representative anatomical motion.

A study established [21] that regardless of treatment modality and delivery properties, the dynamic 4D dose will converge to the 4D dose depending on the number of fractions and the magnitude of target motion. Therefore, interplay effect simulation would still be needed to estimate the dosimetric consequences of moving targets prior to treating certain disease sites with a particular number of fractionations. Notably, the current landscape lacks commercially available treatment planning systems capable of integrating the temporal structure of delivery during optimization. Therefore, the interplay effect simulation holds the utmost importance in addressing this critical gap.

## 5. Conclusions

This study investigated proton-based TLI in a cohort of ten pediatric patients, within the framework of the HAPSAA protocol. Through meticulous analysis, peak-to-peak splenic motion was determined to be within the range of ≤12 mm from 4D CT scans. Notably, interplay effect simulations illuminated that spleen coverage, as dictated by protocol tolerances, remained satisfactory for peak-to-peak splenic motion levels of ≤8 mm. These findings highlight the potential of 4D plan evaluation and robust optimization strategies to mitigate challenges associated with respiratory motion in proton TLI treatments; these strategies are particularly relevant when alternative motion management techniques such as breath-hold are unfeasible due to factors like young age or anesthesia.

Given the absence of published studies detailing the treatment planning intricacies and motion management strategies in pediatric proton-based TLI, this research contributes significantly to the existing body of knowledge. By shedding light on the effectiveness and safety of proton-based TLI in a pediatric cohort, this study’s focus on the spleen, a dynamically contracting and relaxing organ near the diaphragm, offers a valuable step forward in understanding and optimizing treatment outcomes in this specialized context. Our hope is that incorporation of proton-based TLI will reduce normal tissue injury from radiation exposure and improve long-term outcomes of HCT recipients.

## Figures and Tables

**Figure 1 cancers-15-05161-f001:**
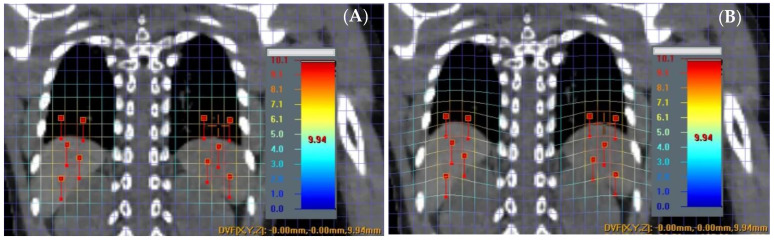
A deformed image of before (**A**) and after (**B**) is shown with 10 anchor points. Cross hair reads a net deformation of 9.94 mm for the spleen in superior direction.

**Figure 2 cancers-15-05161-f002:**
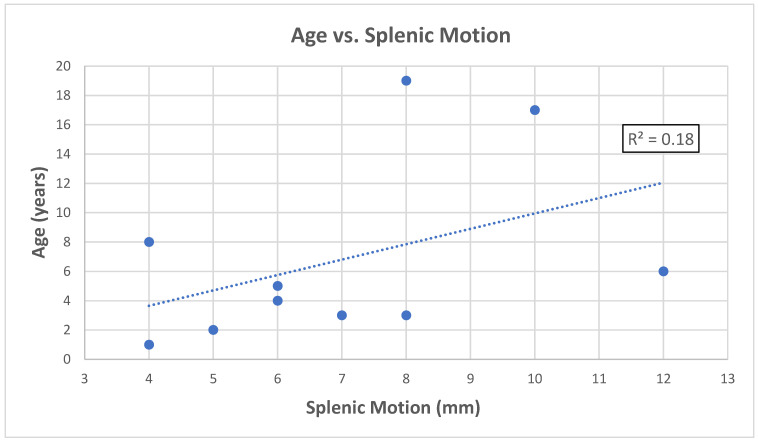
Patients’ age and the magnitude of peak-to-peak splenic motions in inferior-to-superior direction.

**Figure 3 cancers-15-05161-f003:**
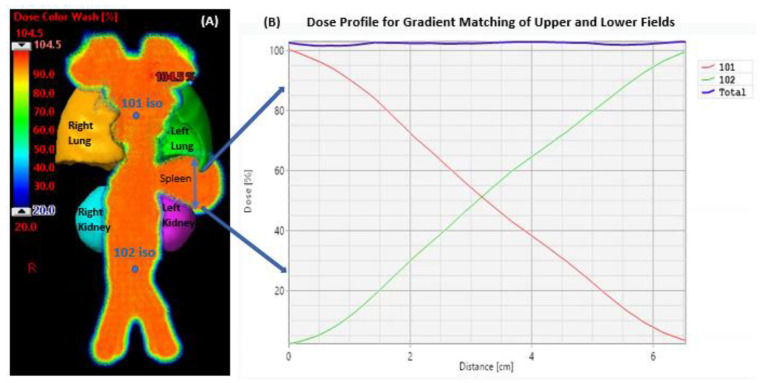
3D TLI dose cloud is shown in color wash painted by two posterior fields (**A**) with an overlap at the mid-spleen for about 6.5 cm revealed in the line dose profile (**B**). Representative plan is from an eight-year-old with 4 mm of superior-to-inferior splenic motion.

**Figure 4 cancers-15-05161-f004:**
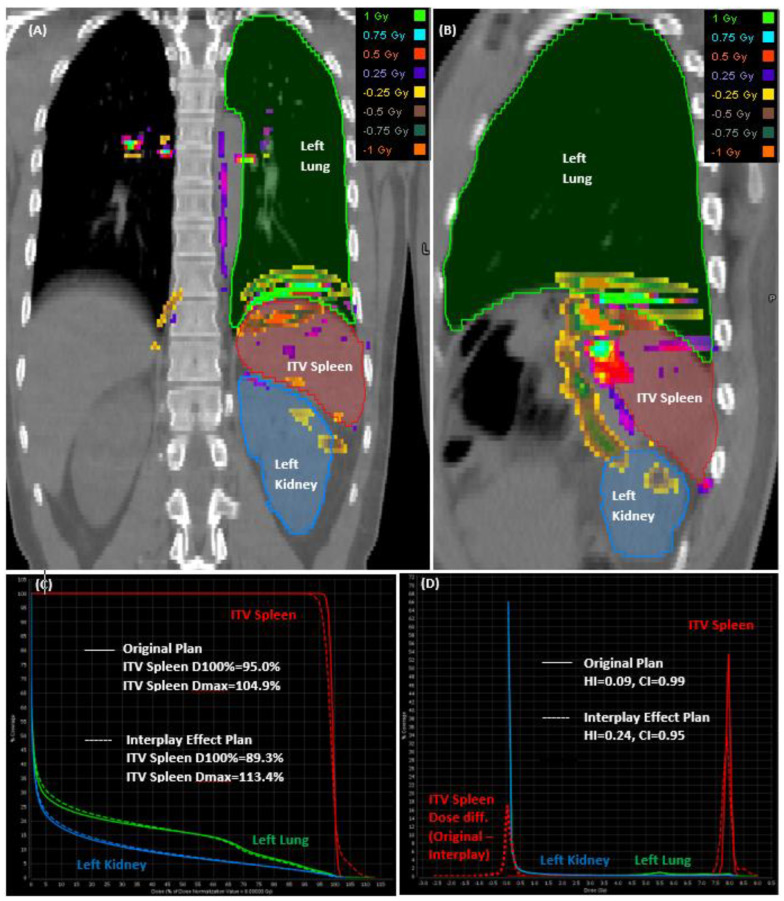
The dose subtraction between original and interplay effect plan seen from the coronal (**A**), and sagittal (**B**) planes, as well as the cumulative (**C**) and differential (**D**) DVHs of patient case #9 with 10 mm of splenic motion.

**Table 1 cancers-15-05161-t001:** The worst-case scenario calculation for splenic motion and its impact on plan quality.

Splenic Motion (mm)	Maximum Exhalation ITV Spleen D100% (%)	Maximum Inhalation ITV Spleen D100% (%)
3	93.9	93.9
6	93.9	92.6
8	75.8	72.9
10	48.7	55.6

**Table 2 cancers-15-05161-t002:** The interplay effect simulation of splenic motion and dose volume parameters of targets and OARs for a cohort of 10 patient cases.

Patient Number	Patient Age (Years)	Breaths per Minute	Peak-to-Peak Splenic Motion (mm)	Original Left Lung/Left Kidney D50% (Gy-RBE)	Interplay Effect Left Lung/Left Kidney D50% (Gy-RBE)	Original Plan ITV TLI Lower/TLI Upper D95% (%)	Interplay Effect ITV TLI Lower/TLI Upper D95% (%)	Original Plan Dose ITV Spleen D100% (%)	Interplay Effect Dose ITV Spleen D100% (%)	Dose Difference ITV Spleen (Original–Interplay) (%) (TH = ±5%)
1	1	31	4	4.7/4.3	4.7/4.4	99.9/100	99.2/93.1	95.5	95.4	0.10
2	8	26	4	0.5/0.3	0.7/0.3	95.3/95.1	92.5/93.8	94.9	92.2	2.70
3	2	17	5	4.8/5.2	4.8/5.2	99.8/97.5	99.2/97.1	94.7	94.8	−0.10
4	5	22	6	2.8/0.7	3.2/0.8	98.7/97.8	97.9/96.8	96.7	98.2	−1.50
5	4	32	6	3.9/7.6	3.7/7.7	101/102.1	101/100.2	95.2	94.3	0.90
6	3	26	7	3.5/2.2	3.5/2.2	99.8/99.9	95.1/98.3	94.8	93.8	1.00
7	19	20	8	1.8/0.2	2.2/0.2	96.6/96.1	95.4/95.3	96.5	95.6	0.90
8	3	24	8	5.2/3.9	5.5/4.0	97.8/98.7	100.1/98.2	95.0	92.2	2.80
9	17	15	10	0.1/0.1	0.1/0.1	95.3/95.1	92.5/93.8	95.0	89.3	5.70
10	6	23	12	2.9/2.8	3.0/2.8	100.7/101	99.2/101.8	98.4	90.6	7.80

## Data Availability

The data that support the findings of this study are available from the corresponding author, upon reasonable request.

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
