# Peer review of "Interplay Effect of Splenic Motion for Total Lymphoid Irradiation in Pediatric Proton Therapy"

_cancers, 2023, doi:10.3390/cancers15215161_

Round 1

Reviewer 1 Report

Comments and Suggestions for Authors

4D dose reconstruction in pencil beam proton therapy is typically based on a single pre-treatment 4DCT. However, splenic motion during fractionated treatment can vary significantly in both amplitude and frequency. In the article, "Interplay Effect of Splenic Motion for Total Lymphoid Irradiation in Pediatric Proton Therapy," the authors investigated the relationship between the magnitude of splenic motion and its effect on plan quality for total lymphoid irradiation. Although the issue of respiratory motion has been extensively reviewed and corrected in various radiotherapy techniques, the approach in this manuscript is relevant to proton therapy dosimetry in pediatric patients with lymphoma, leukemia, and autoimmune disorders. The manuscript is short but concise. It is well organized, with an adequate description of the methodology and convincing results. I believe it can be published as is.

Reviewer 2 Report

Comments and Suggestions for Authors

The topic of this article is of clinical interest.

Comments:

1. The demographic and clinical data of the patients are not clear from the current description. Although it is written that these were pediatric patients, Figure 2 and Table 2 have a patient over 18 years of age. A patient at 19 years of age is an adult patient. The criteria need to be more clearly spelled out. What were the inclusion and exclusion criteria? How were these patients selected?

2.      What does it mean that patient 10 in table 2 had 7 breaths per minute? It's probably a typo.

3. Was spleen motion and size assessed, and was the relationship of spleen motion and size to the patients' clinical findings analyzed? Spleen size depends on age, growth especially in adolescence, and immune diseases, etc.

Reviewer 3 Report

Comments and Suggestions for Authors

This study investigated the proton based TLI in a cohort of ten pediatric patients, within the framework of the HAPSAA protocol.

Through meticulous analysis, peak-to peak splenic motion was determined to be within the range of ≤12 mm from 4D CT scans.

Notably, interplay effect simulations illuminated that spleen coverage, as dictated by protocol tolerances, remained satisfactory for peak-to-peak splenic motion levels of ≤8 mm.

Given the absence of published studies detailing the treatment planning intricacies and motion management strategies in pediatric proton based TLI, this research contributes significantly to the existing body of knowledge.

By shedding light on the effectiveness and safety of proton based TLI in a pediatric cohort, the study's focus on the spleen, a dynamically contracting and relaxing organ near the diaphragm, offers a valuable step forward in understanding and optimizing treatment outcomes in this specialized context.

Our hope is that incorporation of proton-based TLI will reduce normal tissue injury to radiation exposure and improve long term outcomes of HCT recipients.

Round 2

Reviewer 2 Report

Comments and Suggestions for Authors

Although the authors have added new information, questions about the article remain.

Comments:

1.    It is necessary to add references to documents (HAPSAA protocol and others) that define the age of patients under 21 years as pediatric patients, as these data do not correspond to the norms accepted in many countries and may mislead readers. It is necessary to specify in the abstract what was meant by pediatric patients.

2.    Unfortunately, the authors did not add new clinical information about the patients, so it is not possible to determine whether the splenic motion patterns are age- or disease-related.

Round 3

Reviewer 2 Report

Comments and Suggestions for Authors

Although the authors have made corrections, questions remain about the patients' clinical data and how the coefficient of determination R2=0.65 was calculated and what level of statistical significance this model has (Figure 2).

Author Response

Thanks for allowing us to make a correction in the Figure 2. We corrected the linear fit parameter as R2=0.18 and found the correlation to be a weak correlation between patients' age and splenic motion. We also revised the verbiage in the page 4 and line 173-174 in the attached manuscript.

Thank you.

Round 4

Reviewer 2 Report

Comments and Suggestions for Authors

The authors have addressed my concerns about the statistical analysis of the data and have corrected the text of the article accordingly. Unfortunately, there are no clinical data on the patients, which would have improved the quality of the manuscript. However, since the authors no longer refer to data on the relationship between age and spleen motion, this does not seem critical. There are no other comments on the manuscript.